# Passive Monitoring of Parkinson Tremor in Daily Life: A Prototypical Network Approach

**DOI:** 10.3390/s25020366

**Published:** 2025-01-09

**Authors:** Luc J. W. Evers, Yordan P. Raykov, Tom M. Heskes, Jesse H. Krijthe, Bastiaan R. Bloem, Max A. Little

**Affiliations:** 1Center of Expertise for Parkinson and Movement Disorders, Department of Neurology, Donders Institute for Brain, Cognition and Behaviour, Radboud University Medical Center, 6525 GA Nijmegen, The Netherlands; 2Institute for Computing and Information Sciences, Radboud University, 6525 EC Nijmegen, The Netherlands; tom.heskes@ru.nl; 3School of Mathematical Sciences, University of Nottingham, Nottingham NG9 2SE, UK; 4Pattern Recognition & Bioinformatics Group, Delft University of Technology, 2628 XE Delft, The Netherlands; 5School of Computer Science, University of Birmingham, Birmingham B15 2TT, UK

**Keywords:** tremor modeling, tremor detection, passive monitoring, wearable sensors, prototype networks, Parkinson’s disease

## Abstract

Objective and continuous monitoring of Parkinson’s disease (PD) tremor in free-living conditions could benefit both individual patient care and clinical trials, by overcoming the snapshot nature of clinical assessments. To enable robust detection of tremor in the context of limited amounts of labeled training data, we propose to use prototypical networks, which can embed domain expertise about the heterogeneous tremor and non-tremor sub-classes. We evaluated our approach using data from the Parkinson@Home Validation study, including 8 PD patients with tremor, 16 PD patients without tremor, and 24 age-matched controls. We used wrist accelerometer data and synchronous expert video annotations for the presence of tremor, captured during unscripted daily life activities in and around the participants’ own homes. Based on leave-one-subject-out cross-validation, we demonstrate the ability of prototypical networks to capture free-living tremor episodes. Specifically, we demonstrate that prototypical networks can be used to enforce robust performance across domain-informed sub-classes, including different tremor phenotypes and daily life activities.

## 1. Introduction

Parkinson’s disease (PD) has a tremendous impact on the over 6 million people affected by this disease worldwide, and also on their caregivers and other near ones [1]. PD is characterized clinically by a wide range of motor symptoms, such as bradykinesia, rigidity and tremor, and non-motor symptoms, such as sleep problems, depression, autonomic dysfunction, and cognitive impairments. The presence and severity of these symptoms strongly vary, both between patients, due to factors such as disease stage and subtype (e.g., tremor-dominant vs. postural instability and gait difficulties), and within the same patient, due to factors such as stress and response fluctuations in relation to dopaminergic medication use. This makes it challenging to evaluate PD, in particular during short in-clinic visits, which are often not representative of the patient’s condition in real-life [2,3]. In addition, clinical rating scales such as the MDS-UPDRS suffer from intra- and inter-rater variability, which makes it difficult to reliably track within-person changes over time [4].

Unobtrusive wearable sensors offer potentially critical improvements by allowing us to capture objectively and continuously how PD patients function in their daily life [5]. As such, the use of wearable sensors to monitor PD is increasing, both in the (remote) care setting to provide guidance to healthcare providers [6], and in clinical trials to provide more sensitive outcome measures to detect treatment effects of potentially disease-modifying therapies [7]. However, for wearables to deliver on the promise of improving our understanding of free-living PD, we need reliable and interpretable digital biomarkers that reflect how different symptoms evolve over time.

Tremor is an important focus area for sensor-based monitoring of PD because it is highly relevant for early PD patients [8], who form the main target population of trials investigating disease-modifying therapies, and because it is feasible to measure tremor using wearable movement sensors [9]. Tremor is defined as an involuntary, rhythmic, and oscillatory movement of a body part [10]. In PD, rest tremor is the most prevalent subtype, although some patients may also experience re-emergent tremor, i.e., a postural tremor with the same frequency as rest tremor, pure postural tremor, or action tremor [11,12]. The fundamental frequency of rest tremor ranges from 4 to 7 Hz [10]. Rest tremor usually presents in the arms (initially unilaterally), with the exact movement depending on the muscle groups involved. This can vary between and within patients, even on a short timescale (e.g., within minutes). Rest tremor is often an early sign of PD, being one of the presenting signs in 78% of PD patients [13], and progresses during the early stages of the disease [8]. In a large survey, early PD patients most frequently selected tremor as the symptom with the biggest impact on their quality of life [14].

To enable passive monitoring of tremor using wearable sensors, we need models that can reliably detect tremor episodes in the highly variable context of daily life. Attempts to translate algorithms trained in highly controlled environments to passive monitoring in real-life have been largely unsuccessful [15]. Part of the challenge is the real-life heterogeneity of both the tremor class (e.g., with the exact movement depending on the muscle groups involved), and the non-tremor class (with a large variety of real-life behaviors). In the typical supervised machine learning setting, it would require large amounts of representative labeled data to train a classifier for the presence of tremor in real-life conditions. However, labeled datasets collected during unscripted daily life activities, such as the study from Cole et al. [9] and the Parkinson@Home Validation Study [16] have relatively small sample sizes, because such studies are labor-intensive and therefore costly to perform at a large scale.

To address these challenges and develop a robust tremor model, we propose to use *prototypical networks* based on *radial basis functions* (RBFs). This approach allows for the representation of observations in terms of their distances to *prototypical examples*. We select these examples to represent the real-life heterogeneity of both the tremor and the non-tremor classes, based on clinical expertise. Using realistic data from the Parkinson@Home Validation study, we evaluate the ability of the proposed prototypical networks to capture free-living tremor episodes using wrist-worn accelerometers. Specifically, we assess the detection accuracy in the context of limited amounts of labeled training data. In addition, we assess the robustness of our model to different real-life behaviors (which may introduce false-positives), and robustness to different tremor phenotypes (which may introduce false-negatives).

## 2. Related Work

In this section, we discuss prior work focused on detecting and quantifying tremor in persons with PD using wearable sensors. Most studies have trained and evaluated models in standardized settings, often using standardized tasks in highly controlled environments such as the hospital. However, there are an increasing number of studies that include daily life activities, on which we will focus our review.

### 2.1. Sensor Types and Locations

In terms of sensor types, most studies used only accelerometer sensors [17,18,19,20,21,22,23], only gyroscope sensors [24,25], or a combination of both [26,27]. One study focusing on free-living tremor modeling used surface electromyography in combination with accelerometers [28]. Studies commonly used a single sensor device placed at the wrist of the most affected side [17,18,26,27]. Other studies used multiple devices, placed at both wrists [20,21,24]; the wrist and the ankle [25,28]; or both wrists, both legs, waist, and chest [23]. Consumer devices such as smartphones were also used to measure tremor, for example by analyzing the accelerometer signal when patients were reading or writing on the smartphone [19] or when patients were making phone calls [22]. With the aim of achieving high long-term compliance, we opted to use a single device [29]. We focus on using wrist-worn accelerometers, because of their wide availability (e.g., in consumer smartwatches) and because PD tremor usually presents in the upper limbs [8].

### 2.2. Study Designs

To develop and validate models to estimate the presence and severity of tremor from the raw sensor signals, representative labeled datasets are essential. The majority of studies used data collected during scripted activities of daily life (ADLs) performed in the lab, where clinical experts provided the ground truth labeling for the presence and sometimes also the severity of tremor (often based on simultaneous video recordings) [17,18,19,20,23,24,25,27]. Such study designs do not capture the heterogeneity present in real-life conditions, which makes it difficult to predict how models developed using these data will perform in daily life. On the other hand, other studies that do include representative free-living data for an extended period (e.g., continuous passive monitoring for one week), are limited by the available ground truth labels, often consisting of a single clinical evaluation (e.g., the UPDRS tremor items) [17,18,21,22]. Such sparsely labeled data make it difficult to train models for the continuous evaluation of the presence and severity of tremor.

A few studies tried to find the middle ground between representative data on the one hand, and detailed ground truth labels on the other hand. Salarian et al. and Lang et al. included unscripted, patient-initiated activities, with ground truth labels provided by multiple clinical evaluations (e.g., the UPDRS tremor items) [24,26]. However, these activities were still performed in an unnatural environment, i.e., in and around the hospital. Cole et al. included unscripted activities in a simulated home environment (for 4 h), with continuous ground truth labels for the presence and severity of tremor and dyskinesia provided by clinical experts [9,28]. In our work, we use data from the Parkinson@Home Validation Study [16], which employs a similar study design to Cole et al., with the most important difference being that data are collected in and around the patients’ own homes, in order to provide a more accurate reflection of the heterogeneity in free-living data.

### 2.3. Methodology

Many studies used thresholding applied to various frequency-domain features extracted from the windowed accelerometer and/or gyroscope signals (with the window length varying from 1.5 to 30 s) [17,18,21,24]. Thresholds were set by manual adjustment, or minimizing an error function.

In addition, various machine learning approaches have been evaluated to detect tremor in free-living conditions. Rigas et al. (2016) used a decision tree based on window-based frequency-domain features [27]. Cole et al. incorporated the time dependency of consecutive windows, and evaluated a dynamic neural network (DNN), dynamic support vector machine, and a hidden Markov model (HMM), based on window-based frequency- and time-domain features [9,28]. Rigas et al. (2012) also used HMMs to model tremor [23]. Interestingly, in addition to classifying the tremor severity using an HMM, they specifically modeled action versus posture using a separate HMM. Lang et al. used Gaussian processes to model the tremor severity based on various 1 min window-based features (based on wavelet decomposition and power spectral density) [26]. Finally, Hssayeni et al. compared an ensemble model based on gradient tree boosting with a long short-term memory neural network, both using 5 s window-based frequency-domain and cross-correlation features [25].

A few studies specifically developed models to deal with the discrepancy between the time resolution of available labels (e.g., patient self-rated tremor severity) and the desired time resolution of predictions. Papadopoulos et al. combined deep learning and multiple-instance learning, based on window-based frequency-domain features [22]. Zhang et al. developed various weakly supervised variants of support vector machines, neural networks, iterative discriminative axis parallel rectangle, and expectation maximization diverse density, based on window-based frequency-domain features [20].

Currently, the available tremor detection methods often treat PD tremor as one class, or as multiple classes based on the tremor severity. However, this does not reflect the heterogeneity of PD tremor, which can differ both in terms of tremor sub-type (i.e., rest, re-emergent rest, action, or pure postural tremor) and exact tremor movement depending on the muscle groups involved. In addition, the non-tremor class is highly heterogeneous as well, comprising many different daily life activities. To develop a tremor model that is robust to different tremor presentations, in the absence of large amounts of labeled data, we build on the idea of prototype-based learning. This approach allows us to incorporate domain knowledge by labeling well-behaved examples, which represent the heterogeneity in both the tremor and non-tremor classes.

## 3. Materials and Methods

In this section, we propose an intuitive prototypical network approach for constructing a tremor detector with domain-informed prototypical examples of important heterogeneity in the tremor and non-tremor classes. In the current section, we describe and motivate the general approach, whereas in Section 4, we describe how we have trained and evaluated this approach to capture free-living tremor episodes using wrist accelerometers, using data from the Parkinson@Home Validation study.

### 3.1. Prototypical Networks: A Brief Introduction

Prototype-based learning provides an efficient mechanism for summarizing complex datasets and making interpretable inference, particularly in the context of limited amounts of labeled data. Specifically, prototypical layers can be used to embed inductive biases and domain-informed constraints in the architecture of a neural network. This can be achieved using RBF layers [30], which have been popular in early neural network models. In recent years, there has been a renewed interest in using an RBF layer as a final prototypical layer in few-shot learning tasks [31], which can be used to achieve reliable classification performance with only a small number of training samples. However, RBF layers have hardly been used for constructing multi-layer architectures due to the complex inference of their basis parameters and weights. Multi-layer RBF networks are known to exacerbate the vanishing gradient problem associated with the training of DNNs [32,33].

In our work, we address this issue by introducing a novel learning paradigm for multi-layer prototypical RBF networks. First, we define a single-layer prototypical network and describe our approach for incorporating prototypical examples. Then, we extend this model by adding a dimensionality reduction RBF layer, and we describe how learning the weights of this two-layer prototypical RBF network can still be achieved in a tractable way.

### 3.2. Single-Layer Prototypical Network

Here, we propose to use a single-layer prototypical RBF network for the task of detecting free-living tremor episodes.

#### 3.2.1. Model Specification

Let us assume we have a set of input features with a real-valued vector $x_{n}\in R^{D}$ where *D* denotes the input dimensionality and n∈{1,…,N} indexes the *n*-th data point. The RBF networks define a map $\varphi :R^{D}\to R$, which can be viewed as a superposition of a set of *K* basis functions:(1)$$\varphi \left(x_{n}\right)=\sum_{k=1}^{K}\omega_{k}\rho \left(d\left(x_{n},c_{k}\right)\right)$$
where $\left\{\omega_{1},\ldots ,\omega_{K}\right\}$ denote inferable weights, $\left\{c_{1},\ldots ,c_{K}\right\}$ denote the basis parameters, which we will use to encode additional information via prototypical examples, $\rho \left(\cdot \right)$ is a pre-defined basis that is associated with a non-linearity such as a Gaussian function or polyharmonic spline [34], and $d\left(\cdot \right)$ is any valid distance function such as the Euclidean distance. Depending on the shape of the non-linearity and the distance function, the bases can be parametrized with a varying number of parameters, e.g., basis centers and scales. In this work, we have used the Gaussian basis $\rho \left(\cdot \right)$, which is bounded in the range $\left(0,1\right)$, and the Mahalanobis distance $d\left(\cdot \right)$.

The outputs from Equation (Equation 1) are passed through a standard softmax layer to output the binary tremor/non-tremor classification.

#### 3.2.2. Model Training

We start training the RBF network by inferring the basis parameters $\left\{c_{1},\ldots ,c_{K}\right\}$ from prototypical examples. Based on domain knowledge, we fix the number of Λ sub-classes in the different *L* classification categories. In our case, L=2 since we are interested in tremor/non-tremor classification, and Λ represents the number of unique tremor and non-tremor sub-classes. We assume that, from each of the Λ sub-classes, annotated prototypical examples are available, i.e., representative and diverse examples of the feature vectors $\left\{x_{1},\ldots ,x_{N}\right\}$ (in Section 4, we describe how the prototypical examples were obtained in our study). For each sub-class $\lambda \in \left\{1,\ldots ,\Lambda \right\}$, we estimate multiple basis parameters $c_{k}$, by summarizing the multi-modal prototypical data (as displayed in Section 4) with a flexible Dirichlet process mixture (DPM) [35], trained using scalable maximum a posteriori inference [36]. (We use a Gaussian DPM with diagonal Normal inverse Gamma prior placed on the component means and variances. The DPM hyperparameters are inferred in an unsupervised manner using standard Bayesian model selection, by maximizing the complete data log-likelihood.) The inferred component parameters of all Λ mixture models are then taken to be the basis parameters (i.e., basis centers and scales). As a consequence, while Λ and *L* are fixed based on domain knowledge, the total number of bases *K* is modeled as a random variable since each λ-sub-class DPM infers an unknown $K_{\left(\lambda \right)}$ number of components, i.e., the total number of bases is the sum of the random variables $K=K_{\left(1\right)}+\cdots +K_{\left(\Lambda \right)}$.

Once the basis parameters are estimated, the second step of training the RBF networks involves fitting a linear model with coefficients $\omega_{1},\ldots ,\omega_{K}$ to the outputs with respect to an objective function. For our binary tremor/non-tremor classification task, we have used a logistic model.

### 3.3. Two-Layer Prototypical Network

A limitation of single-layer prototypical networks is that their discriminatory power depends on the relative distances to prototypes in a potentially cumbersome high-dimensional feature space. For example, for high-dimensional $x_{n}\in R^{D}$ with Gaussian basis $\rho \left(\cdot \right)$ and Euclidean distance $d\left(\cdot \right)$, the distances and the corresponding basis outputs become diluted. This has motivated the use of hierarchical architectures, where supervised training is specified in a lower-dimensional embedding [37]. A widely used version of prototypical layers that augment existing embedding architectures is the prototypical network from Snell et al. [31]. In this prototypical network, domain-informed RBF activation functions are used in the final supervised layer. The approach we take in this work is similarly motivated, but we show how prototypical RBF layers can be used to construct a two-layer network.

#### 3.3.1. Model Specification

In our proposed two-layer prototypical RBF network, the input features $x_{n}\in R^{D}$ are parsed through a dimensionality reduction RBF layer before predicting the class probabilities. We denote the first layer map with $\varphi^{\left(1\right)}:R^{D}\to R^{M}$, such that:(2)$$\varphi^{\left(1\right)}\left(x_{n}\right)={\left[\sum_{k=1}^{K^{\left(1\right)}}\omega_{k,m}^{\left(1\right)}\rho^{\left(1\right)}\left(d^{\left(1\right)}\left(x_{n},c_{k}^{\left(1\right)}\right)\right)\right]}_{m=1}^{M}$$
where $c_{1}^{\left(1\right)},\ldots ,c_{K^{\left(1\right)}}^{\left(1\right)}$ are the basis parameters used for the dimensionality reduction of the input features. In our experiments below, we have used D=45 for the original feature space and M=15 for the intermediate embedding space.

The second layer predicts the class probabilities as in Section 3.2, based on the distance to the prototype bases, i.e., the second layer map is $\varphi^{\left(2\right)}:R^{M}\to R$ with:(3)$$\varphi^{\left(2\right)}\left(x_{n}^{*}\right)=\sum_{k=1}^{K^{\left(2\right)}}\omega_{k}^{\left(2\right)}\rho^{\left(2\right)}\left(d^{\left(2\right)}\left(x_{n}^{*},c_{k}^{\left(2\right)}\right)\right)$$
with $x_{n}^{*}\in R^{M}$ denoting an intermediate embedding of $x_{n}$. $\rho^{\left(1\right)}\left(\cdot \right)$ and $\rho^{\left(2\right)}\left(\cdot \right)$ are layer-specific bases, $d^{\left(1\right)}\left(\cdot \right)$ and $d^{\left(2\right)}\left(\cdot \right)$ are layer-specific distance functions, and $\omega_{k,m}^{\left(1\right)}$ and $\omega_{k}^{\left(2\right)}$ are layer-specific weights. Figure 1 provides a graphical representation of the two-layer prototypical RBF network.

#### 3.3.2. Model Learning

The second layer basis parameters $c^{\left(2\right)}$ are set using Gaussian DPMs as in Section 3.2.2, based on the original feature space $R^{D}$, but mapped onto $R^{M}$ by the updated $\varphi^{\left(1\right)}\left(x_{n}\right)$.

To set the bases in the first layer $c_{1}^{\left(1\right)},\ldots ,c_{K^{\left(1\right)}}^{\left(1\right)}$, we leverage the link between RBF networks and Gaussian processes, and use the *informative vector machines* framework [38,39,40] for selecting $K^{\left(1\right)}$
*inducing points* to act as both training inputs and basis centers (i.e., the basis scales are fixed to a small constant). The inducing points are latent (i.e., lower-dimensional embeddings) variables optimized to preserve the structure of the inferred $\varphi^{\left(1\right)}\left(\cdot \right)$ as if it was trained using all *N* data points. The general motivation for this is that the first layer reduces the dimensionality of the data, whereas the second layer encodes domain knowledge through prototypical examples. In this work, the number of inducing points was inferred using standard grid search and leave-one-patient-out cross-validation metrics.

Multi-layer RBF networks are known to exacerbate the vanishing gradient problem associated with the training of DNNs [32,33]. We tackle this issue by altering conventional back-propagation. Specifically, we propose to optimize the weights via an intermediate step, which optimizes the embedding $x_{n}^{*}$.

We assume a standard cross-entropy loss of the two-layer RBF network, because of the specific binary classification goal:(4)$$E=-\frac{1}{N}\sum_{n=1}^{N}\left[y_{n}\log\left({\hat{y}}_{n}\right)+\left(1-y_{n}\right)\log\left(1-{\hat{y}}_{n}\right)\right]$$
where we have used $y_{n}\in \left\{0,1\right\}$ to indicate if the true label is associated with an observation *n*; ${\hat{y}}_{n}$ indicates the predicted class terms (more specifically, ${\hat{y}}_{n}$ are obtained by applying the softmax layer to the network outputs $\varphi^{\left(2\right)}(\varphi^{\left(1\right)}\left(x_{n}\right))$).

Instead of propagating the updates of the weights $W^{\left(1\right)}=\left\{\omega_{1}^{\left(1\right)},\ldots ,\omega_{K_{1}}^{\left(1\right)}\right\}$ and $W^{\left(2\right)}=\left\{\omega_{1}^{\left(2\right)},\ldots ,\omega_{K_{2}}^{\left(2\right)}\right\}$ from the gradients of the loss function *E*, we optimize the desired internal embeddings $X^{*}=\left\{x_{1}^{*},\ldots ,x_{N}^{*}\right\}$ with respect to the loss (note that the complexity of the internal embeddings is regularized by the fact that they are a linear lower-dimensional projection of $\rho^{\left(2\right)}\left(d^{\left(2\right)}(X^{*},C^{\left(2\right)})\right)$ specified via the weights matrix $W^{\left(1\right)}$ [41,42]).

First, the internal embeddings are updated in the direction of the gradient:(5)$$X^{*}=X^{*}-\eta \frac{\partial E}{\partial X^{*}}$$
where η denotes the learning rate. The gradient of the loss function can be derived via the chain rule:(6)$$\begin{array}{cc} \frac{\partial E}{\partial X^{*}} & =\frac{\partial E}{\partial \Phi^{\left(2\right)}}\frac{\partial \Phi^{\left(2\right)}}{\partial X^{*}}=\\ \; & =\frac{\partial E}{\partial \Phi^{\left(2\right)}}\frac{\partial \rho^{\left(2\right)}\left(d^{\left(2\right)}(X^{*},C^{\left(2\right)})\right)}{\partial X^{*}} \end{array}$$
where we have used the matrix notation $\Phi^{\left(2\right)}\in R^{N\times K}$ to indicate the distance of each point to each of the second layer bases $C^{\left(2\right)}=c_{1}^{\left(2\right)},\ldots ,c_{K^{\left(2\right)}}^{\left(2\right)}$ (i.e., $\Phi^{\left(2\right)}=\rho^{\left(2\right)}\left(d^{\left(2\right)}\left(X^{*},C^{\left(2\right)}\right)\right)$. Similarly, $\Phi^{\left(1\right)}\in R^{N\times K}$ indicates the distance of each point in *X* to each of the first layer bases $C^{\left(1\right)}=c_{1}^{\left(1\right)},\ldots ,c_{K^{\left(1\right)}}^{\left(1\right)}$. Note that in Equation (Equation 6), the loss function *E* also depends on the weights $W^{\left(2\right)}$, but we do not explicitly include the partial derivative $\frac{\partial E}{\partial W^{\left(2\right)}}$ since $X^{*}$ is treated as inferable and $W^{\left(2\right)}$ is considered deterministic (i.e., given optimized $X^{*}$) as will be shown next.

Second, given updated $X^{*}$, the first layer weights $W^{\left(1\right)}$ are recomputed such that they provide a least-squares fit to $X^{*}$, using the pseudo-inverse (i.e., Moore–Penrose inverse [43]):(7)$$W^{\left(1\right)}={\left(\Phi^{\left(1\right)}\right)}^{-1}X^{*}.$$

Third, given the updated first layer weights $W^{\left(1\right)}$, we recompute $X^{*}$ and $\Phi^{\left(2\right)}$. Using the updated $\Phi^{\left(2\right)}$, we then apply a gradient update to the second layer weights $W^{\left(2\right)}$. Under the binary cross-entropy loss, this takes the form:(8)$$W^{\left(2\right)}=W^{\left(2\right)}-\eta \frac{1}{N}{\left(\Phi^{\left(2\right)}\right)}^{T}\left(\hat{Y}-Y\right)$$The proposed training method utilizes standard optimization for the second layer weights $W^{\left(2\right)}$.

These three steps are performed iteratively until convergence is reached (with the stopping criterion based on the change in the loss function). The first layer weights $W^{\left(1\right)}$ are initialized to be the eigenvectors projecting each $x_{n}$ from $R^{D}\to R^{M}$ (i.e., the principal components estimated using the inducing points preserving the covariance of *X*). The second layer weights $W^{\left(2\right)}$ are randomly initialized using a standard normal distribution.

Approaching the inference problem via optimization of the internal embeddings, rather than the network weights directly, allows us to regularize the weights updates (reducing the curvature of the trained models [44]). This strategy was proposed in the context of RBF networks in Lowe and Tipping [44], and later extended for Gaussian process latent variable models (GPLVMs) in Lawrence [42] and deep Gaussian process models in Damianou and Lawrence [45].

## 4. Experimental Evaluation

Here, we evaluate the ability of the proposed prototypical networks to capture free-living tremor episodes among PD patients using wrist-worn accelerometers. In this section, we describe the dataset used, how we labeled the prototypical examples, how we pre-processed the accelerometer data, and how we evaluated the performance of the single- and two-layer prototypical networks from Section 3.

### 4.1. Dataset

We used data from the Parkinson@Home Validation study. A detailed description of the study design can be found in Evers et al. [16]. In brief, the study included 25 PD patients with motor fluctuations and 25 age-matched, non-PD controls. During visits to the participants’ own homes, unscripted daily life activities and standardized clinical assessments (including the MDS-UPDRS part III) were recorded on video while the participants wore various wearable sensors. Participants in the PD group were recorded twice: once after overnight withdrawal of dopaminergic medication and again starting from 1 h after medication intake. The Parkinson@Home Validation study was approved by the local medical ethics committee (Commissie Mensgebonden Onderzoek, regio Arnhem-Nijmegen, file number 2015–1776). Informed consent was obtained from all participants prior to enrollment.

In this work, we used the raw accelerometer data collected from the wrist-worn Physilog 4 devices during the unscripted activities. From the PD patients with tremor, we used the data from the arm with the most severe tremor (based on the MDS-UPDRS part III before medication intake, items 3.15 and 3.17). From the other participants, we selected the sides to match for hand dominance. Due to technical problems with the sensor devices of one PD patient and one non-PD control, we included 24 PD patients and 24 controls in the analyses. Table 1 provides more information about the demographic and clinical characteristics of the study sample.

Based on the video recordings, trained research assistants annotated the main activities (e.g., walking, sitting, standing still) and symptoms (e.g., tremor and freezing of gait) occurring during the unscripted activities. The symptom annotations were checked by a movement disorders expert. The annotators did not have access to the participants’ clinical data. In this work, we used the annotations for tremor presence in the arm with the most severe tremor (i.e., the same side as the accelerometer sensor). In 8 PD patients, tremor was observed during the unscripted daily life activities, whereas in the other 16 PD patients, no tremor was observed. The flowchart for the tremor annotations of the 8 PD patients with tremor is presented in Figure 2. In summary, if the patient performed significant upper limb activities for more than 3 s, the assistant only annotated whether tremor was present or not. Otherwise, both the presence and severity of the tremor were annotated, similar to the MDS-UPDRS part III tremor items. However, because of the low prevalence of moderate and severe tremor in this dataset, we aimed to model only the presence of tremor.

In addition to the continuous annotations for the presence and severity of tremor, the video recordings of all PD patients with tremor were also screened to select prototypical examples of different tremor sub-classes. These examples were to used to infer the basis parameters in our prototype model (see Section 3). After consultation with a neurologist specialized in Parkinson tremor (Rick Helmich, MD, PhD), we defined seven tremor sub-classes based on the dominant tremor movement: (1) wrist and/or fingers flexion–extension, while the arm is supported; (2) wrist and/or fingers flexion–extension, while the arm is free; (3) elbow flexion–extension, while the arm is supported; (4) elbow flexion-extension, while the arm is free; (5) pronation-supination of the lower arm, while the arm is supported; (6) pronation–supination of the lower arm, while the arm is free; and (7) tremor during gait. The tremor sub-types were specifically defined for the annotation of free-living video recordings; we choose to focus on different visible movements depending on the muscle groups involved in the tremor, and not explicitly on the different tremor sub-types (i.e., rest, re-emergent rest, pure postural, and action tremor), because it can be difficult to distinguish between these sub-types based on free-living video recordings. Figure 3 visualizes the spectrograms of the accelerometer data collected during the unscripted activities of the 8 PD patients with tremor, while Figure 4 highlights annotated prototypical examples of the different tremor sub-classes.

Similarly to selecting examples for the tremor class, we also defined sub-classes to represent the heterogeneous non-tremor class. For this, we used the video annotations for the main activities in combination with the tremor annotations. We defined the following seven non-tremor sub-classes: (1) gait (i.e., when the participant walks five or more consecutive steps); (2) postural transitions (i.e., when the participant is transitioning between sitting, standing, or laying); (3) running or performing sport exercises; (4) driving a bike or car; (5) sitting, standing still, or laying down, while performing “suspicious” upper limb activities (i.e., periodic activities of the hand or arm that could be misclassified as tremor), (6) sitting, standing still, or laying down, while performing other upper limb activities (all activities except for the activities referred to in category 5); and (7) sitting, standing still, or laying down, without any significant upper limb activities. The non-tremor sub-classes are exhaustive and mutually exclusive, i.e., all non-tremor observations are associated with one of these 7 sub-classes, with the exception of subjects without tremor annotations; because the tremor annotations were needed to distinguish between sub-classes 5, 6, and 7, these sub-classes were not available from these subjects. During model training, all available data in each of the non-tremor sub-classes are used for the DPM-based clustering to infer the basis parameters (see Section 3).

### 4.2. Pre-Processing and Feature Extraction

First, we down-sample the raw three-axial accelerometer data from the Physilog devices from 200 Hz to 50 Hz after anti-aliasing with a fourth-order moving average filter. To remove the effect of orientation changes of the device, we apply $l_{1}$-trend filtering to each individual axis, assuming piecewise linear changes [46,47] (i.e., setting λ to 10,000).

In order to avoid spectral estimation artefacts such as *Gibbs phenomenon* when extracting window-based features, we first fit a Bayesian non-parametric switching autoregressive model [47] to the signal magnitude that estimates the boundaries of approximately stationary segments. Within these stationary segments, 15 different features are extracted from each axis of the pre-processed accelerometer data (resulting in a total of 45 features for all axes combined), based on non-overlapping 2 s windows: the standard deviation; the power in different frequency bands (0.3–2 Hz, 4–8 Hz, 8–12 Hz, 0.2–14 Hz); the frequency and height of the dominant peak in the PSD within different frequency bands (0.3–2 Hz, 4–8 Hz, 8–12 Hz, 0.2–14 Hz); the sample entropy; and the spectral entropy. We then perform z-score normalization of the feature vectors, using the data from the unscripted activities of all 24 PD patients (both with and without annotated tremor episodes) and all 24 non-PD controls.

### 4.3. Model Evaluation

We evaluate the ability of the proposed prototype models to capture free-living tremor episodes among PD patients using wrist-worn accelerometers. The evaluation compares the single-layer and two-layer RBF prototypical networks, and two baseline models: a random forest classifier and a logistic classifier.

All methods are trained and evaluated using leave-one-subject-out cross-validation (CV). During training, all models have access to the data from all subjects except for one. Data from the remaining subject are then used for testing. This applies to both data from the prototypical examples (used to infer the basis parameters in the prototype models) and the remaining data. To avoid a high number of false-positives given the higher prevalence of the non-tremor class, during training all models are optimized for sensitivity at specificity in the range of (0.945, 0.955). The logistic classifier is $l_{2}$ regularized. For both baseline models, we correct for the tremor/non-tremor class imbalance (i.e., the weights of the logistic classifier are re-weighted using the relative class frequencies in the training data; for the random forest standard re-sampling using bootstrap is adopted).

During testing, we evaluate the models’ sensitivity, specificity, and AUROC using the same thresholds set during training. In addition, we assess the robustness of our model to different real-life behaviors (which may introduce false-positives), and robustness to different tremor phenotypes (which may introduce false-negatives). We do this by computing the sensitivity stratified for different tremor sub-classes, and specificity stratified for different non-tremor sub-classes. Lastly, to assess the performance of the prototype models in the context of limited amounts of labeled training data, we compute the learning curves showing how the AUROCs change with increasing amounts of training data. We hypothesize that the proposed prototype models can achieve acceptable performance with a minimal amount of training data.

Due to the limited sample size and the rare occurrence of specific sub-classes, conducting statistical tests to rigorously compare the performance of different models was not feasible. Therefore, we instead opted for a qualitative comparison of the models’ performance and robustness, with the aim to obtain meaningful insights into the differences between models.

## 5. Results

In this section, we evaluate the ability of the proposed prototypical networks to detect tremor episodes in free-living conditions, using the data and evaluation procedures described in Section 4.

### 5.1. Tremor Detection Performance

In Table 2, we report the tremor detection performance of all models measured in terms of sensitivity, specificity, and AUROC. With respect to the average performance, the proposed single-layer prototype model performed similarly to the logistic classifier. The two-layer prototype model demonstrated a higher mean sensitivity and AUROC in comparison to both the single-layer version and the baseline methods. In addition, the variability in the sensitivity and AUROC across patients was lower for both prototype models compared to the baseline methods.

### 5.2. Robustness Across Activities and Tremor Sub-Classes

In Table 3, we report the specificity stratified for the different non-tremor activities. The most notable difference between the methods is the higher average specificity during certain rarer sub-classes, i.e., “Driving bike/car”, “Upper limb activities”, and “Suspicious activities”, for both prototype models compared to the baseline methods, which is associated with a lower variance across patients.

In Table 4, we report the sensitivity stratified for different tremor sub-classes. Both prototype models demonstrated a higher sensitivity for five out of seven tremor sub-classes, associated with a lower variance across patients, and a similar sensitivity for the other two tremor sub-classes. The higher sensitivity is most notable for the tremor during gait sub-class, of which only 20% (random forest classifier) or 56% (logistic classifier) of observations are correctly classified as tremor by the baseline models, whereas the sensitivity for the prototype models is 80% for the two-layer version and 84% for the single-layer version.

### 5.3. Learning Curves

In Figure 5, we show the learning curves for both prototype models and both baseline methods. When the models had access to only a limited amount of labeled training data (i.e., 2 min or less), both prototype models demonstrated a higher AUROC in comparison to the baseline models. When having access to at least 20 min of training data, differences between the different models become smaller, although the mean AUROC of the two-layer prototype model is still slightly higher when using the full dataset (as reported in Table 2). In contrast, the single-layer prototype model appears to be insufficiently flexible to learn from increasing amounts of training data, resulting in a slightly lower mean performance when trained on the full dataset.

### 5.4. Robustness Across Patients

In Table 5, we delve into the individual performance of the two-layer prototype model of all PD patients with tremor (n = 8). The AUROC across all PD patients with tremor is 0.80 or higher, with the exception of one patient (PD_4) with an AUROC of 0.75. Regarding specificity, we see that five out of eight PD patients have a specificity of 0.94 or higher. The lower specificities in PD_3 (0.87) and PD_8 (0.86) are accompanied by high AUROCs of 0.90 or higher, which suggests that personalized thresholds would have been more optimal for these subjects. The individual sensitivities vary from 0.43 to 0.90. As expected, the variation in sensitivity is partly explained by each individual’s tremor severity; a higher rest tremor severity of the arm of the most affected side (measured by the MDS-UPDRS item 3.17, on and off states combined) is associated with a higher sensitivity (Pearson R of 0.66, 80% confidence interval of 0.21 to 0.88; see Figure 6). Last, there is a high absolute agreement between the predicted and true total time with tremor (intra-class correlation (ICC) of 0.80, 80% confidence interval of 0.55 to 0.92; see Figure 7).

## 6. Discussion

We have studied the problem of detecting tremor episodes in free-living conditions using wrist accelerometer data, and proposed a modeling approach that can incorporate domain knowledge into RBF networks by using prototypical examples. Using a realistic dataset that incorporates unscripted daily life activities, we showed that prototypical RBF networks possess some useful properties when used to detect tremor among PD patients. Although the single-layer version showed a similar average performance to the baseline models, the two-layer prototypical RBF network demonstrated a higher discriminative ability, which was associated with a lower variability across patients. In addition, we showed that the prototype models can be used to enforce robust performance across domain-informed sub-classes, including specific daily life activities and tremor presentations. Lastly, the learning curves suggest that the prototype models can provide acceptable performance with limited amounts of labeled training data, which is a useful property given the scarcity of labeled free-living data.

### 6.1. Comparison to Other Approaches

The proposed prototype models possess some useful properties, which go beyond the reported empirical performance. We use prototypical examples to construct compact, generative density models, which allow for the estimation of uncertainty associated with our predictions. In addition, the estimated confidence of the prototype models depends on the distance of the samples from the training distribution, which is a useful property for out-of-distribution detection [48]. In the Bayesian setting, the proposed models specify a locally stationary process around the prototype locations. The approach can be seen as inferring a GP covariance, only using the similarities to prototypical examples as explanatory variables, rather than all raw signal features. The practical advantage of that is that we can account for the real-life heterogeneity of tremor and non-tremor in a tractable manner. Whereas most existing tremor detection methods treat PD tremor as one class [5], the proposed models offer a natural way to incorporate existing knowledge about different tremor presentations in free-living conditions. In addition, the non-tremor class is highly heterogeneous as well, comprising many different daily life activities. Although it may not be feasible to capture examples of all possible daily life activities, the prototype models can be used to enforce robustness to common daily life activities, or rhythmic daily life activities, which are likely to be misclassified as tremor [19,49] (such as walking or riding a bike).

### 6.2. Limitations

The findings of this study should be interpreted in light of its limitations. Firstly, the number of PD patients experiencing tremor included in the Parkinson@Home Validation Study was limited (n = 8). Because of this, in combination with the rare occurrence of specific tremor presentations and activities, it was not feasible to conduct statistical tests to rigorously compare the performance of the different models. Therefore, the generalizability of the reported differences in performance should be confirmed on an independent dataset, preferably including more PD patients who experience tremor. This evaluation should preferably also include a comparison with other published models for tremor detection. Secondly, the prototypical examples used in the current evaluation should be considered as a starting point. In this work, we demonstrate a proof-of-concept of prototype models using different visible movements depending on the muscle groups involved in the tremor. We envision that the current selection could be further extended to also incorporate examples of different tremor sub-types, i.e., rest, re-emergent rest, action, or pure postural tremor [11,12]. In the current work, we did not focus on distinguishing between these different tremor sub-types because the required annotations were not available from the video recordings of the Parkinson@Home Validation Study. With a further extension of the different tremor presentations, it may become relevant to not only detect tremor, but also quantify the association with different tremor sub-classes, which is something the prototype models can do by design. This could be particularly relevant for the management of tremor since available treatment options differ between tremor sub-types [50]. Thirdly, the proposed models do not yet exploit the time-dependent nature of tremor and human behavior. Other studies have investigated the use of dynamic neural networks, hidden Markov models, and dynamic support vector machines for this purpose [9,27,28]. Future work could embed the proposed prototype models in sequential latent variable models such as hidden Markov models with piecewise non-linear states, e.g., adopting an RBF network hidden Markov model (RBF-HMM) [51,52]. The potential benefits of such a compositional representation would not only be to further improve tremor detection performance, but also to infer common sequential predictors of tremor episodes.

### 6.3. Future Directions

Objective and continuous insights into the presence of tremor in free-living conditions could benefit both individual patient care and clinical trials, by overcoming the snapshot nature of clinical assessments [53]. The proposed prototype models can contribute to both applications by allowing for reliable detection of tremor episodes, but more work remains necessary to capitalize on the expected benefits.

The management of PD tremor involves different treatment options, including both pharmacological interventions such as dopaminergic and anti-cholinergic medication, and non-pharmacological interventions such as deep-brain stimulation and stress-reducing strategies [50,54]. It varies between patients which treatment is most effective, depending on, e.g., the tremor sub-type, the sensitivity to stress, and the presence of dopamine-resistant tremor [50,55]. Insights into the individual course of PD tremor in daily life, and the effects of stress and intake of dopaminergic medication, could allow for more informed personalized treatment decisions, and for accurate follow-up of the treatment response. Although the number of available remote monitoring systems for PD is increasing [5], the added value for the management of tremor—in terms of its impact on clinical decision-making, quality of life, and functioning of patients—remains to be elucidated.

In the context of clinical trials, reliable passive monitoring of tremor could offer more sensitive outcome measures to evaluate the effects of potential disease-modifying therapies [56]. Tremor is a promising measurement target because (1) it is present in early PD, which is the target population of most disease-modification trials [8,13], and (2) it is an important aspect to monitor from the perspective of patients [14,57]. The use of sensor-based outcome measures in clinical trials is increasing [58,59]. However, the focus in clinical trials has largely been on active monitoring, i.e., measuring the execution of structured tasks. The proposed prototype models allow for the continuous, passive monitoring of tremor in daily life, which also reduces the required effort from participants and is associated with excellent long-term compliance [29]. Future work will assess the suitability of the proposed models to capture long-term disease progression, fueled by the emergence of large cohort studies that include passive monitoring using a wrist-worn sensor device, such as the Parkinson Progression Marker Initiative and the Personalized Parkinson Project [60,61].

## Figures and Tables

**Figure 1 sensors-25-00366-f001:**
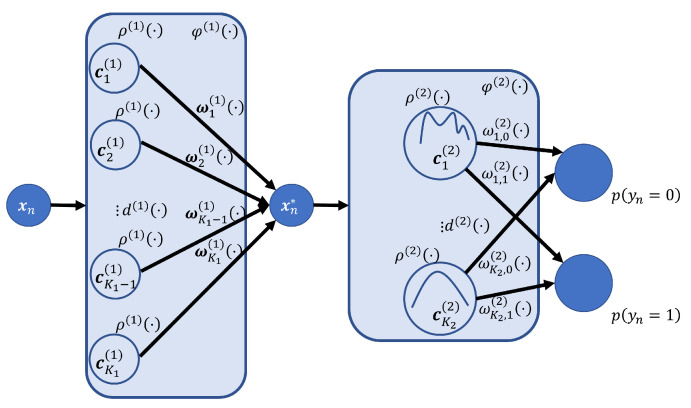
An illustration of the two-layer prototype model. Each input $x_{n}$ is first reduced to $R^{M}$ by the layer $\varphi^{\left(1\right)}$ (Equation (Equation 2)) and then to a class probability via layer $\varphi^{\left(2\right)}$ (Equation (Equation 3)), summarized in the shaded boxes.

**Figure 2 sensors-25-00366-f002:**
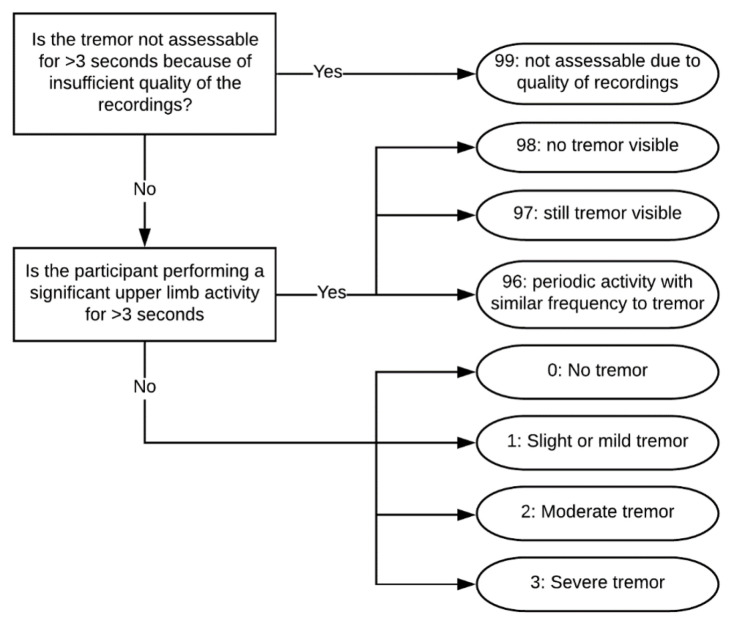
Schematic overview of the video annotation protocol for the presence and severity of tremor used in the Parkinson@Home Validation study.

**Figure 3 sensors-25-00366-f003:**
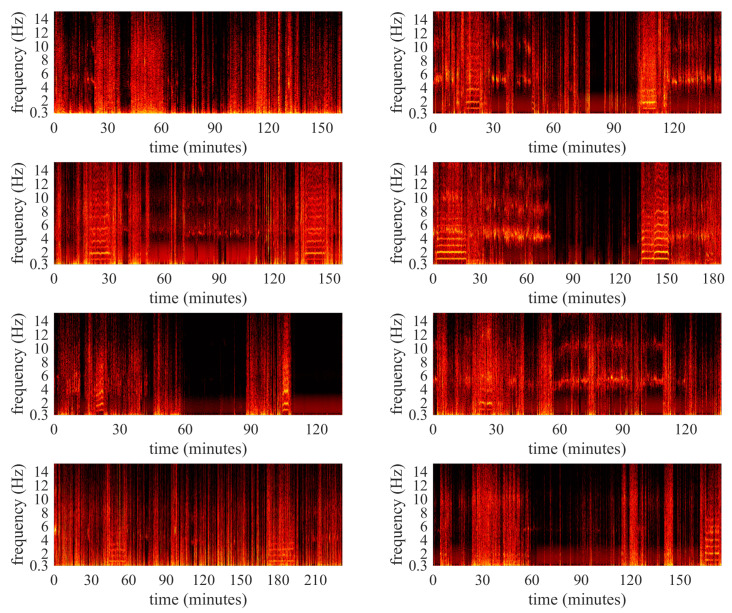
Spectrograms of the *x*-axis of the accelerometer data collected during the unscripted activities of the 8 PD patients with tremor (each panel represents one patient). Higher brightness denotes higher spectral power in the corresponding frequency bins. Welch’s method is used to average over periods of 20 s (2 s windows with 50% overlap).

**Figure 4 sensors-25-00366-f004:**
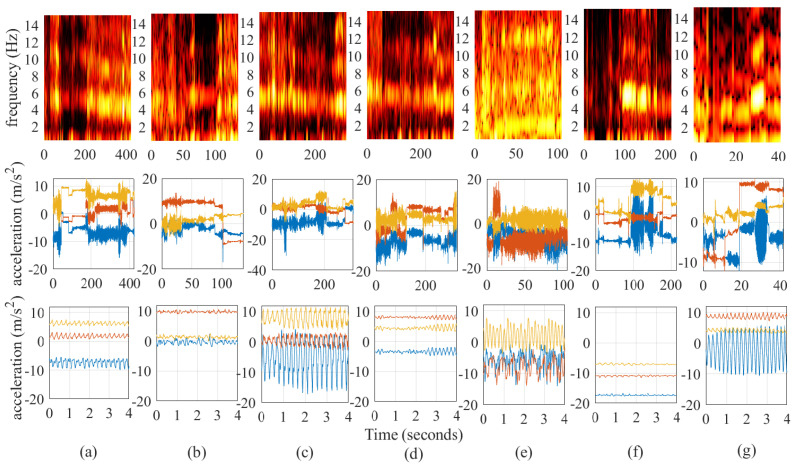
Visualization of the annotated prototypical examples for the different tremor sub-classes. The top panels display the spectrograms of all data from the prototypical examples from each tremor sub-class. Higher brightness denotes higher spectral power in the corresponding frequency bins. Welch’s method is used to average over periods of 20 s (2 s windows with 50% overlap). The middle panels display the corresponding raw accelerometer data (blue: x-axis, red: y-axis, yellow: z-axis). The bottom panel zooms in on stationary segments from the prototypical examples of each tremor sub-class. The tremor sub-classes are denoted as follows: (**a**) wrist and/or fingers flexion–extension, while the arm is supported; (**b**) wrist and/or fingers flexion–extension, while the arm is free; (**c**) elbow flexion–extension, while the arm is supported; (**d**) elbow flexion–extension, while the arm is free; (**e**) tremor during gait (**f**) pronation–supination of the lower arm, while the arm is supported; and (**g**) pronation–supination of the lower arm, while the arm is free.

**Figure 5 sensors-25-00366-f005:**
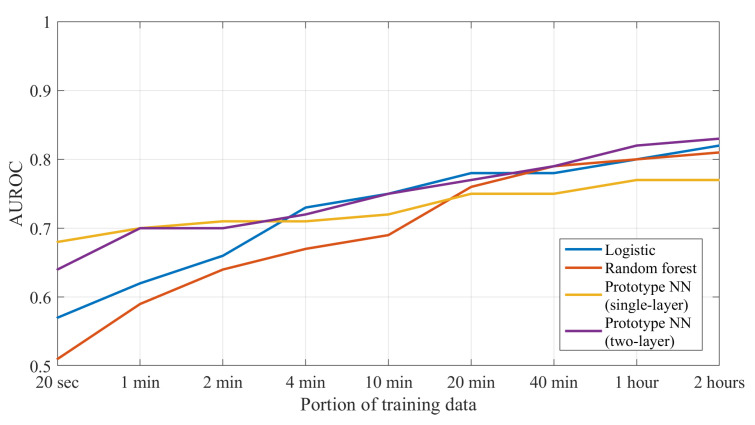
Learning curves of the prototype models and baseline models, showing the models’ performance with increasing duration of training data. The lines indicate the average AUROC across folds, evaluated using leave-one-subject-out cross-validation across the 8 PD patients with annotated tremor episodes. As we extend the duration of training data (i.e., as we move from the left to the right on the plot), we add new training data to the existing training data sample, to reduce the effect of random sampling when comparing different training data durations.

**Figure 6 sensors-25-00366-f006:**
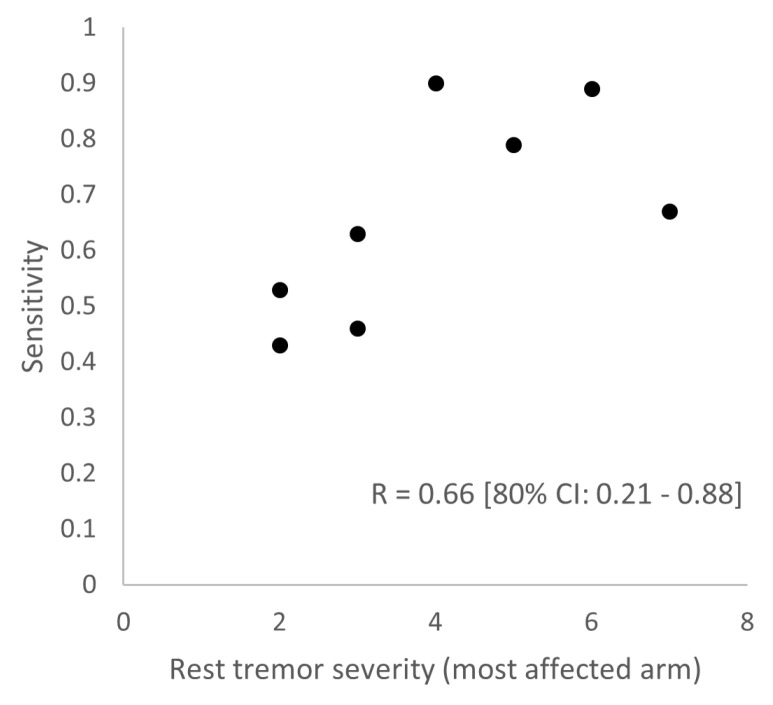
Correlation (Pearson R) between the individual sensitivity of the two-layer prototype model and the rest tremor severity of the most affected arm (same side as the device), according to the MDS-UPDRS item 3.17 (sum of measurements in the on and off states). 80% CI: 80% confidence interval.

**Figure 7 sensors-25-00366-f007:**
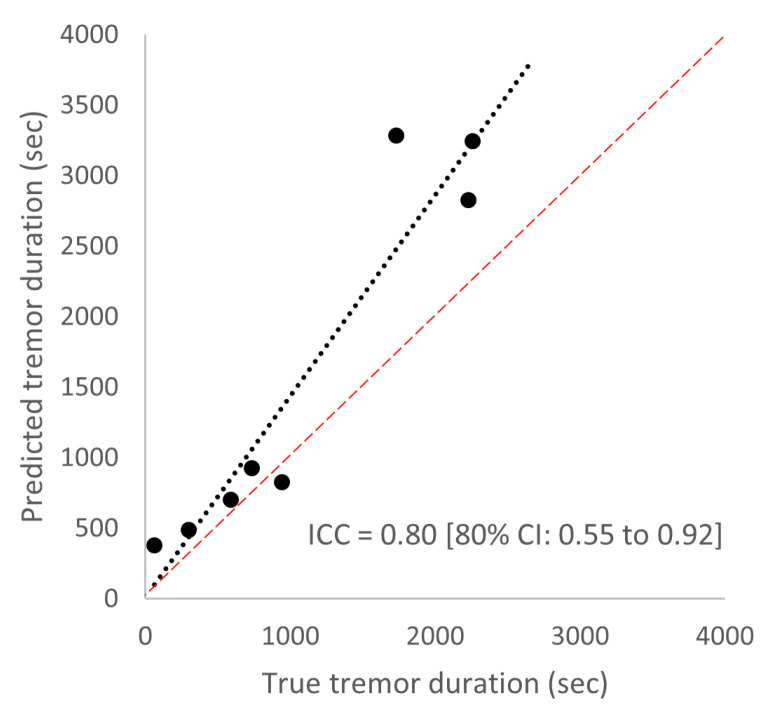
Agreement between the true tremor duration (according to the video annotations) and predicted tremor duration (according to the two-layer prototype model), with the red line indicating perfect agreement. ICC: intra-class correlation, 80% CI: 80% confidence interval.

**Table 1 sensors-25-00366-t001:** Demographic and clinical characteristics of PD patients and non-PD controls included in the analyses. IQR: inter-quartile range. MDS-UPDRS: Movement Disorder Society-Sponsored Revision of the Unified Parkinson’s Disease Rating Scale. Part 1: non-motor experiences of daily living. Part 2: motor experiences of daily living. Part 3: motor examination. Part 4: motor complications. *: 1 missing value.

	PD Patients with Tremor	PD Patients Without Tremor	Non-PD Controls
	(n=8)	(n=16)	(n=24)
Age (years),	61.0	66.0	67.5
median (IQR)	(58.3–69.0)	(61.0–70.5)	(55.0–70.0)
Gender (men), *n* (%)	4 (50%)	7 (44%)	13 (54%)
Time since diagnosis of PD (years), median (IQR)	7.0 (5.3–9.5)	7.0 (5.0–11.0)	-
Hoehn and Yahr stage in off state, *n* (%)			
Stage 1	1 (12.5%)	0 (0.0%) *	-
Stage 2	6 (75.0%)	10 (62.5%)	-
Stage 3	0 (0.0%)	4 (25.0%)	-
Stage 4	1 (12.5%)	1 (6.3%)	-
MDS-UPDRS			
Part 1 (0–52),	10.5	9.5	3.0
median (IQR)	(7.0–17.3)	(8.0–15.0)	(0.3–4.0)
Part 2 (0–52),	10.5	9.0	0.0
median (IQR)	(8.3–14.5)	(7.3–13.0)	(0.0–0.0) *
Part 3 (off state, 0–132),	50.4	35.0	6.5
median (IQR)	(38.8–61.8)	(30.0–46.8)	(4.3–11.0)
Part 3 (on state, 0–132),	32.0	25.5	-
median (IQR)	(24.0–37.5)	(17.5–38.0)	-
Part 4 (0–24),	6.0	6.0	-
median (IQR)	(3.5–9.8)	(4.3–8.5)	-
Tremor sub-score of MDS-UPDRS part III			
Off state (0–40),	14.0	4.0	0.5
median (IQR)	(10.3–18.8)	(2.3–8.5)	(0.0–1.8)
On state (0–40),	9.5	2.0	-
median (IQR)	(3.0–11.8)	(1.0–5.0)	-
Rest tremor severity (arm of most affected side),			
*n* (%)			
0: normal (off |	0 (0.0%) |	14 (87.5%) |	24 (100%)
on)	2 (25.0%)	15 (93.8%)	
1: slight (off |	1 (12.5%) |	0 (0.0%) |	0 (0.0%)
on)	2 (25.0%)	1 (6.3%)	
2: mild (off |	3 (37.5%) |	1 (6.3%) |	0 (0.0%)
on)	2 (25.0%)	0 (0.0%)	
3: moderate (off |	3 (37.5%) |	1 (6.3%) |	0 (0.0%)
on)	2 (25.0%)	0 (0.0%)	
4: severe (off |	1 (12.5%) |	0 (0.0%) |	0 (0.0%)
on)	0 (0.0%)	0 (0.0%)	

**Table 2 sensors-25-00366-t002:** Tremor detection performance (average across leave-one-subject-out cross-validation folds, and standard deviation between brackets) of the prototype models and baseline methods. The reported sensitivity and AUROC are computed across the 8 PD patients with annotated tremor episodes. The specificity is computed across all 24 PD patients and 24 non-PD controls.

Method	Sensitivity	Specificity	AUROC
Logistic classifier	57% (24%)	95% (6%)	83% (8%)
Random forest classifier	41% (23%)	95% (4%)	81% (9%)
Prototype network (single-layer)	53% (20%)	95% (4%)	83% (6%)
Prototype network (two-layer)	66% (18%)	95% (4%)	87% (7%)

**Table 3 sensors-25-00366-t003:** Specificity stratified for the different non-tremor sub-classes, evaluated using leave-one-subject-out cross-validation across 24 PD patients and 24 non-PD controls. We report the average performance across folds with the standard deviation between brackets. The number of patients with data from the different sub-classes is indicated between brackets. Annotations for the sub-classes “Upper limb activities”, “No upper limb activities”, and “Suspicious activities” were only available from the 8 PD patients with tremor (hence the maximum prevalence of these sub-classes is eight).

Method	Gait $\left(n=48\right)$	Postural Transitions $\left(n=48\right)$	Running/ Exercising $\left(n=5\right)$	Driving Bike/Car $\left(n=16\right)$	Upper Limb Activities $\left(n=8\right)$	No Upper Limb Activities $\left(n=8\right)$	“Suspicious” Activities $\left(n=4\right)$
Logistic classifier	96% (5%)	98% (4%)	95% (7%)	56% (23%)	88% (11%)	92% (6%)	83% (29%)
Random forest	98% (2%)	98% (4%)	87% (14%)	84% (14%)	89% (9%)	90% (8%)	82% (20%)
Prototype network (single-layer)	96% (4%)	96% (6%)	90% (8%)	94% (8%)	93% (5%)	93% (5%)	92% (8%)
Prototype network (two-layer)	96% (4%)	96% (3%)	95% (5%)	89% (6%)	91% (6%)	94% (4%)	93% (8%)

**Table 4 sensors-25-00366-t004:** Sensitivity stratified for the different tremor sub-classes, evaluated using leave-one-subject-out cross-validation across the 8 PD patients with annotated tremor episodes. We report the average performance across folds with the standard deviation between brackets. This evaluation is based on the annotation of prototypical examples of each tremor sub-class. The tremor sub-classes are mutually exclusive but non-exhaustive (i.e., only a part of the tremor observations are labeled with one of the tremor sub-classes).

Method	Wrist/Fingers Flexion–Extension, Arm Supported $\left(n=5\right)$	Wrist/Fingers Flexion–Extension, Arm Free $\left(n=4\right)$	Elbow Flexion–Extension, Arm Supported $\left(n=4\right)$	Elbow Flexion–Extension, Arm Free $\left(n=4\right)$	Tremor During Gait $\left(n=2\right)$	Pro-Supination, Arm Supported $\left(n=3\right)$	Pro-Supination, Arm Free $\left(n=3\right)$
Logistic classifier	64% (30%)	72% (20%)	94% (3%)	97% (3%)	56% (16%)	78% (25%)	56% (13%)
Random forest	64% (32%)	66% (22%)	93% (4%)	96% (4%)	20% (28%)	84% (20%)	52% (26%)
Prototype network (single-layer)	94% (6%)	90% (6%)	94% (5%)	96% (4%)	84% (10%)	97% (6%)	86% (10%)
Prototype network (two-layer)	94% (4%)	92% (3%)	94% (3%)	96% (5%)	80% (16%)	90% (8%)	88% (9%)

**Table 5 sensors-25-00366-t005:** Tremor detection performance of the two-layer prototype model for each PD patient with tremor (n = 8). AUROC: Area under the receiver operator curve. Rest tremor severity of the most affected arm is evaluated using the MDS-UPDRS item 3.17.

Patient ID	AUROC	Sensitivity	Specificity	Amount of Detected Tremor (s)	Amount of Labeled Tremor (s)	Rest Tremor Severity (Off State)	Rest Tremor Severity (On State)
PD_1	0.80	0.43	0.95	934	942	2	0
PD_2	0.83	0.46	0.95	492	298	3	0
PD_3	0.95	0.90	0.87	2832	2228	2	2
PD_4	0.75	0.79	0.67	3286	1730	3	2
PD_5	0.86	0.53	0.94	705	588	1	1
PD_6	0.91	0.67	0.97	930	730	4	3
PD_7	0.92	0.63	0.97	384	60	2	1
PD_8	0.90	0.89	0.86	3246	2256	3	3

## Data Availability

Access to the data from the Parkinson@Home Validation study can be arranged through a request to the Michael J. Fox Foundation (www.michaeljfox.org). For this work, the dataset was accessed on 21 November 2021. Code implementation of the one- and two-layer prototype models is available via https://github.com/JordanRaykov/Prototype-Networks accessed on 21 November 2021.

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
