# Peer review of "Passive Monitoring of Parkinson Tremor in Daily Life: A Prototypical Network Approach"

_sensors, 2025, doi:10.3390/s25020366_

Round 1

Reviewer 1 Report

Comments and Suggestions for Authors

The Authors present a novel approach to the detection of tremor in Parkinson's Disease based on prototypical networks, from data of the Parkinson@Home Validation study.

The paper is well organised, the contents are clearly presented and the topics are well described. The results highlight the significance of the novelty in this topic, and the Authors thoroughly list and discuss both the limitations of the approach and the implications of the present study.

Author Response

We thank the reviewer for the positive and encouraging feedback on our work.

Reviewer 2 Report

Comments and Suggestions for Authors

Overall I think this manuscript has the potential to provide a valuable contribution to the cited literature. However, it's length, level of methodologic detail, and inconsistent (or lack of) description of model outcomes detract from it's practical interpretation. The goal is to classify the presence, type, and duration of tremor with limited training data, but that message is lost or not conveyed in the methods and results. Models are frequently compared, but it is not described how they are compared. Are they compared qualitatively (i.e. AUROC > < = AUROC) or quantitatively (i.e. with DeLong or bootstrap methods). The latter definitely needs to be addressed in the language throughout the manuscript before acceptance. Ideally the clarity and focus of the manuscript would be improved prior to acceptance as well.

A few specific criticisms for the authors:

1. The formatting of Table 1 with the dash at the end of a column prior to moving to the next row is very confusing, as the next column starts immediately adjacent and it's hard to tell on first blush which number the dash precedes. Recommend putting the entire IQR on the next line under the median. Or include some borders.

2. Were "trained research assistants" and the "movement disorders expert" blinded to other study details at the time of their assessments? Knowledge of participants (medication usage pattern, patient-reported outcomes, etc.) could easily bias subjective assessments. This should be described.

3. Is there any scientific or expert consensus-based publication that could justify the 7 tremor subtypes? If so it should be cited, if not some language (a sentence) describing the rationale for the choices should be included.

4. Section 5.3: Not clear from the text that this section is referring to tremor detection and not non-tremor. 

5. Figure 5: There are no R or p-values attached to this figure. Without them, and with limited N (=8), this figure is uninformative.

6. Figure 7: This figure is very important for the overall concept of the paper, that prototypical networks can perform well with limited training data. However this figure is only mentioned once in the text and only with a 2 sentence paragraph. If you have enough training data, such as was contained in this dataset, the models appear qualitatively equivalent.

Author Response

Comment 1: Overall I think this manuscript has the potential to provide a valuable contribution to the cited literature. However, it's length, level of methodologic detail, and inconsistent (or lack of) description of model outcomes detract from it's practical interpretation. The goal is to classify the presence, type, and duration of tremor with limited training data, but that message is lost or not conveyed in the methods and results. Models are frequently compared, but it is not described how they are compared. Are they compared qualitatively (i.e. AUROC > < = AUROC) or quantitatively (i.e. with DeLong or bootstrap methods). The latter definitely needs to be addressed in the language throughout the manuscript before acceptance. Ideally the clarity and focus of the manuscript would be improved prior to acceptance as well.

Response 1: We appreciate your recognition of the potential contribution of our work, and we understand your concerns regarding the clarity and focus of the manuscript. We have shortened the methods section, to reduce redundancy and focus on the core objectives and findings. Regarding the model comparisons, we indeed opted for qualitative comparisons of model performance rather than conducting statistical tests. Because of the small number of subjects, we used a leave-one-subject-out cross-validation (LOSO-CV) scheme to efficiently use the available amount of data for training and evaluation. However, this validation scheme complicates statistical tests for the observed differences. Using individual hold-out subjects' performances from LOSO-CV for statistical tests, such as the DeLong test, risks overestimating the variance across subjects. This arises from the variability introduced by the differences in models trained on slightly different training sets in each fold. In addition, given the small number of subjects, we believe it is in any case difficult to make reliable claims about statistical significance based on this dataset. The current results should be interpreted as an indication that the prototype models can be used to reduce misclassifications of specific sub-classes, also when applied to patients not included in the training set. We agree that if larger datasets become available (in particular with more patients with tremor), statistical tests should be done to confirm the generalizability of the observed differences. We have added a paragraph in the methods section to make clear that in this work, we only qualitatively compared the mean performance across folds, and we have added the considerations about statistical significance to the limitation section of the manuscript. 

Comment 2: The formatting of Table 1 with the dash at the end of a column prior to moving to the next row is very confusing, as the next column starts immediately adjacent and it's hard to tell on first blush which number the dash precedes. Recommend putting the entire IQR on the next line under the median. Or include some borders.

Response 2: Thank you for pointing us to this formatting issue. We have now changed Table 1 to include IQR values on the following line to improve readability (for this update, we have not used track changes). 

Comment 3:  Were "trained research assistants" and the "movement disorders expert" blinded to other study details at the time of their assessments? Knowledge of participants (medication usage pattern, patient-reported outcomes, etc.) could easily bias subjective assessments. This should be described.

Response 3: Thank you for highlighting this important point regarding potential bias in the annotations conducted by the research assistant and neurologist. The annotators indeed only had access to the video recordings, and no other clinical or patient-reported data was available (the clinical tremor rating was used to determine which side should be annotated for the presence of tremor, but this selection was made by another researcher and not by the annotators). We have clarified this in the manuscript.

Comment 4: Is there any scientific or expert consensus-based publication that could justify the 7 tremor subtypes? If so it should be cited, if not some language (a sentence) describing the rationale for the choices should be included.

Response 4: We understand the need for a justification of the chosen tremor sub-types. The 7 tremor sub-types were specifically defined for the annotation of free-living video recordings, in close collaboration with Dr. Rick Helmich, neurologist and internationally recognized expert on Parkinson tremor. The need to define these subtypes, instead of using existing tremor classifications, arose from the free-living nature of the recordings: we choose to focus on different visible movements depending on the muscle groups involved in the tremor, and not explicitly on the pathophysiological tremor subtypes (i.e. rest, re-emergent rest, pure postural or action tremor), because it is difficult to reliably distinguish between these subtypes based on free-living video recordings. We have clarified our rationale for constructing the subtypes in the manuscript.

As we also mention in our discussion, the prototypical examples used in the current evaluation should be considered as a starting point. In this work, we demonstrate a proof-of-concept of prototype models using different visible movements depending on the muscle groups involved in the tremor, as this data was available from the Parkinson@Home Validation Study dataset. If labeled free-living data becomes available which reliably distinguishes between more different tremor subtypes, these could be easily incorporated into the proposed approach. 

Comment 5: Section 5.3: Not clear from the text that this section is referring to tremor detection and not non-tremor. 

Response 5: Thank you for pointing this out - we have clarified in the manuscript (Section 5.3) that this section investigates the individual performance of patients with tremor.

Comment 6: Figure 5: There are no R or p-values attached to this figure. Without them, and with limited N (=8), this figure is uninformative.

Response 6: Thank you for pointing this out - we had only mentioned the Pearson R value in the text, and we have now also added it to Figure 5. Similarly, we have also added the level of agreement (ICC) in Figure 6.

Comment 7: Figure 7: This figure is very important for the overall concept of the paper, that prototypical networks can perform well with limited training data. However this figure is only mentioned once in the text and only with a 2 sentence paragraph. If you have enough training data, such as was contained in this dataset, the models appear qualitatively equivalent.

Response 7: Figure 7 is indeed important to support the conclusion that our proposed model can perform well with small amounts of labeled data, and we understand that it needs more explanation in the results section, in particular in relation to the overall performance reported in Table II-IV. We agree that the main differences between the models is visible for very small amounts of labeled data (<20 mins), although the overall AUROC of the two-layer prototype model is still slightly higher when using the full dataset (as reported in Table II), with the caveat that the statistical significance of this result needs to be confirmed on datasets with more patients with tremor (related to your first comment). However, the overall accuracy is not all that matters: robustness to specific rarer sub-classes of tremor and non-tremor might even be more important for practical applications, as misclassifications of certain activities such as biking can lead to biased estimates when monitoring individual patients (e.g. when evaluating the effect of exercise on tremor, when some patients choose biking as exercise). As shown in Table III and IV, the prototype models provide a way to enforce robustness to specific sub-classes given limited training data. We have expanded the interpretation of Figure 7 in the results section, and placed it in context of the other results.

Reviewer 3 Report

Comments and Suggestions for Authors

This work proposes the use of a prototype network, which can be embedded with domain expertise about heterogeneous tremor and non-tremor sub-classes. And this prototype network can be used in different domains, and also demonstrates the robustness of the prototype network in cross-domain applications, including different tremor phenotypes and activities of daily living. It is recommended to accept and use for publication after addressing the following questions.

1. Is the number of subjects in group 24 too small? It is recommended that the number of subjects be increased to validate accuracy.

2. The work appears to be missing the final conclusions section, please provide them.

3. The authors mention that In a typical supervised machine learning environment, a large amount of representative labelled data is required to train a classifier to determine whether tremor exists in real-life situations. However, labelled datasets collected during unscripted activities of daily living, such as the Cole et al. study and the Parkinson's@Home validation study, have relatively small sample sizes. This is because such studies are labour-intensive. There is nothing here that essentially points to the reliability of small sample data studies, merely a critique of the fact that machine learning requires large amounts of labelled data to train classifiers, and suggests that the introduction section of the manuscript be carefully revised to add credibility and readability to the paper.

4.The work is considered as a research subject, verifying the need to provide relevant ethical certificates.

Author Response

Comment 1: is the number of subjects in group 24 too small? It is recommended that the number of subjects be increased to validate accuracy.

Response 1: We agree that the number of subjects in the available dataset (24 PD patients and 24 controls) is relatively small. On the one hand, it would indeed be beneficial to collect representative labeled datasets at a larger scale. On the other hand, significantly expanding the amount of labeled data is difficult because collecting and labelling data using such a protocol is labor-intensive and costly, and therefore difficult to scale up. Therefore, in this work, we aimed to contribute to this problem by developing a machine learning approach which can learn with limited amounts of training data. In the discussion, we also acknowledge the need to validate our findings on a larger sample size, in particular in a study which includes more PD patients who experience tremor.

Comment 2: The work appears to be missing the final conclusions section, please provide them.

Response 2: In our manuscript, we have integrated the conclusions into the beginning of the discussion section, in line with the Sensors author guidelines, which state that a separate conclusion section is not mandatory unless the discussion is unusually long or complex. Given that our discussion is relatively concise, we felt this was appropriate to avoid unnecessary repetition. However, if the reviewer or editor prefer a separate conclusion section, we would be happy to add it to the manuscript.

Comment 3: The authors mention that “In a typical supervised machine learning environment, a large amount of representative labelled data is required to train a classifier to determine whether tremor exists in real-life situations. However, labelled datasets collected during unscripted activities of daily living, such as the Cole et al. study and the Parkinson's@Home validation study, have relatively small sample sizes. This is because such studies are labour-intensive.” There is nothing here that essentially points to the reliability of small sample data studies, merely a critique of the fact that machine learning requires large amounts of labelled data to train classifiers, and suggests that the introduction section of the manuscript be carefully revised to add credibility and readability to the paper.

Response 3: In the introduction, we intended to highlight the key challenge addressed in this work: both tremor and non-tremor classes are heterogeneous, requiring sufficiently flexible models to capture this variability. However, large-scale collection of labeled data to train such flexible models (without overfitting) is difficult in free-living conditions. Therefore we propose an approach which can incorporate domain knowledge by learning from strategically choosing labeled examples, thereby reducing the need for large amounts of labeled data.

Comment 4: The work is considered as a research subject, verifying the need to provide relevant ethical certificates.

Response 4: Thank you for pointing this out - we have attached the ethical documents for review, and we have mentioned the ethical approval in the manuscript (under “Informed Consent Statement”).

Round 2

Reviewer 3 Report

Comments and Suggestions for Authors

The authors have made one-to-one revisions in response to reviewer‘s comments and recommend acceptance in its current form for publication.